# A co-evolutionary perspective on humans and *Mycobacterium tuberculosis* in the era of systems biology

**Michaela T Reichmann[1], Liku B Tezera[1,2], Laura Denney[1], Hannah Schiff[1], Andres Vallejo[1,2], Salah Mansour[1,2], Alasdair Leslie[3,4,5], Diana J Garay-Baquero[1,2], Paul T Elkington[1,2]***

[1]NIHR Biomedical Research Centre, Clinical and Experimental Sciences Academic Unit, Faculty of Medicine, University of Southampton, Southampton, United Kingdom; [2]Institute for Life Sciences, Southampton, United Kingdom; [3]Africa Health Research Institute, Durban, South Africa; [4]College of Health Sciences, School of Laboratory Medicine and Medical Sciences, University of KwaZulu Natal, Durban, South Africa; [5]Department of Infection and Immunity, University College London, London, United Kingdom

## eLife Assessment

This Review Article explores the intricate relationship between humans and *Mycobacterium tuberculosis* (Mtb), providing an additional perspective on tuberculosis (TB) disease. Specifically, this review focuses on the utilization of systems-level approaches to study TB, while highlighting challenges in the frameworks used to identify the relevant immunologic signals that may explain the clinical spectrum of disease. The work could be further enhanced by better defining key terms that anchor the review, such as 'unified mechanism' and 'immunological route'. This review will be of interest to immunologists as well as those interested in evolution and host-pathogen interactions.

***For correspondence:**
p.elkington@soton.ac.uk

**Abstract** Tuberculosis is once again the most fatal global infectious disease and has killed many more humans than any other pathogen. Despite the identification of *Mycobacterium tuberculosis* (Mtb) over 140 years ago, we have yet to control the epidemic. A central issue is the complexity of the host–pathogen interaction, with multiple underlying pathways leading to tuberculosis disease. This intricate relationship stems from the prolonged co-evolution of the pathogen with humans, resulting in diverse immunological processes leading to tuberculosis disease. Conversely, Mtb exposure may give a survival advantage through innate immune training, thereby providing selective pressure over millennia. Emerging methodologies, such as single-cell and spatial transcriptomics, offer a golden opportunity to understand the immunology unpinning this host–pathogen interaction at unprecedented resolution. However, these analyses will be fundamentally flawed if they do not consider the intricacies of human Mtb infection. Here, we propose that attempts to find single immunological mechanisms leading to tuberculosis are hindering progress, and we must embrace the complexity of multiple paths to disease to allow the systems biology era to deliver transformative solutions.

## Introduction

The study of tuberculosis (TB) pioneered infectious disease research in the modern scientific era, contributing to the formulation of Koch's postulates demonstrating that an illness can have an infectious origin (*Kaufmann, 2003*). *Mycobacterium tuberculosis* (Mtb) infects and survives within macrophages, subverting the host immune response by multiple mechanisms including inhibition of phagosome maturation and downregulation of antigen presenting molecules, leading to the formation of complex immune aggregates, known as granulomas (*O'Garra et al., 2013*). Even though TB was the first definitively identified infection, it remains the world's deadliest infectious disease despite well over 100 years of research (*Trajman et al., 2025*). This contrast raises a critical question: why is Mtb proving so resistant to human efforts to control it?

Mtb has eluded attempts to develop a fully protective vaccine, despite a partially effective vaccine being available since the 1920s. *Mycobacterium bovis* BCG was developed by sequential culture of *M. bovis* and protects children against disseminated TB but has limited protection against adult disease (*Mangtani et al., 2014*; *Trunz et al., 2006*). Although exciting progress has been made with a vaccine that reduces progression to overt TB disease by 50% when given to those with immunological evidence of latent infection in a phase II study (*Tait et al., 2019*), currently the immune mechanisms underpinning protection have not been identified. Subsequently, a major trial of an alternative vaccine showed no efficacy in preventing recurrence. Indeed, despite being highly immunogenic, the relapse rate tended to be higher in the vaccine group (*Borges et al., 2025*). Similarly, repeat BCG vaccination does not increase protection, despite inducing a strong Mtb-specific CD4 T cell response (*Schmidt et al., 2025*).

These difficulties highlight the priority of understanding the host–pathogen interaction more fully. We have insufficient knowledge of key steps in disease progression to develop transformative interventions. Heterogeneity across the spectrum of human TB is well described (*Barry et al., 2009*; *Cadena et al., 2017*), but the majority of fundamental investigations into disease mechanisms are based on the premise of a consistent underlying process, and that this can be understood through reductionist scientific approaches. However, clinical observations demonstrate that there are multiple paths to TB, and so seeking to define a single mechanism is likely to be flawed.

## Immunological insights from historic and recent clinical observations

Mtb has co-evolved with humans for millennia, with some estimates suggesting up to 70,000 years (*Brites and Gagneux, 2015*), though other analyses suggest the most recent common ancestor was ~6000 years before present (*Bos et al., 2014*; *Kay et al., 2015*). The field of paleoarchaeology provides extensive evidence of TB from the early Neolithic period in the Middle East, with approximately 5% of skeletons from a 10,000-year-old village showing signs of TB (*Dutour, 2023*). This was just before animal domestication and pottery, in hunter-gatherers who built stone houses, and so Mtb was already successfully transmitting in humans before the subsequent population growth that occurred with farming (*Dutour, 2023*). Potentially, to survive in relatively small hunter-gatherer communities, Mtb may have needed to have reduced virulence to avoid excessive deaths and a latent period to permit sustainable transmission in low population numbers (*Gagneux, 2012*). With the expansion and increased density of human populations, more rapid progression to TB disease can be sustained, consistent with analysis that most TB progression in high-incidence settings occurs within 1–2 years of exposure (*Behr et al., 2018*).

Mtb successfully persisted over the ages and then flourished in the crowded populations that occurred with the industrial revolution (*Dubos and Dubos, 1987*), giving rise to the modern TB era approximately 250 years ago. The fundamental cause of TB remained unknown until Koch's seminal work (*Kaufmann, 2003*). Early investigations demonstrated that the first Mtb infection point was the lung base, while Mtb exits from the apices of the lungs (*Ghon, 1916*). This life cycle must involve several distinct host/pathogen interactions, as initially immune evasion is required for Mtb to survive, but then later immune engagement is necessary to cause the inflammation and lung destruction needed to optimize transmission (*Elkington and Friedland, 2015*). Cavitary lung disease leads to proliferation of extracellular bacteria and increased transmission (*Yoder et al., 2004*). Notably, most people (approximately 90%) initially infected with Mtb never progress to active, clinical disease (*Trajman et al., 2025*).

In addition, in the pre-antibiotic era, the progression and regression of different lesions in the same individual were observed on chest radiographs, and one third of patients with active TB disease self-healed, showing that the host–pathogen interaction is finely balanced at all stages of infection (*Dubos and Dubos, 1987*).

Modern immunological techniques and the development of biologic therapies that target specific immune processes have provided extensive insight into TB disease mechanisms. The greatly increased occurrence of TB in the context of HIV co-infection, for example, highlighted immunodeficiency as a major driver of disease (*Kwan and Ernst, 2011*). Similarly, the occurrence of TB after anti-TNF-α therapy for autoimmune disease confirmed the importance of TNF-α in control of latent infection (*Keane et al., 2001*). Furthermore, genetic investigations have identified numerous immunodeficiencies via studies of Mendelian Susceptibility to Mycobacterial Diseases (MSMD), with mutations typically along the IL-12/IFN-γ/STAT signaling pathway (*Jouanguy et al., 1996*; *Dupuis et al., 2001*; *Arias et al., 2024*). With less clearly defined immunologic mechanisms, malnutrition is a significant risk factor for TB (*Dheda et al., 2016*), and food supplementation reduces TB incidence in contacts (*Bhargava et al., 2023*). Therefore, diverse immune deficiencies can lead to active TB.

The vast majority of patients who develop TB, however, have no clear identifiable immunodeficiency. Indeed, Comstock's seminal study from the 1970s showed that children from Haiti with the strongest recall responses to Mtb antigens actually had the greatest subsequent risk of developing TB (*Comstock et al., 1974*). These observations have been replicated in more recent studies using IFN-γ release assays (IGRAs) in response to TB antigens, where higher IFN-γ production associates with increased risk of progressing to disease in both children and adults (*Andrews et al., 2017*; *Ledesma et al., 2021*). TB is most common in young adults in their immunological prime and more frequent in males than females, characterized by an excessive inflammatory response (*Horton et al., 2016*). The implication that TB can also be caused by immune excess is now supported by recently introduced cancer immunotherapies (*Tezera et al., 2020b*). Anti-PD-1 treatment, which activates the immune response and represents the immunological opposite to anti-TNF therapy, should control TB if immunodeficiency were the critical component. However, anti-PD-1 treatment can lead to rapid reactivation of latent TB infection, first identified in case reports (*Elkington et al., 2018*). This finding is supported by studies in mice (*Lázár-Molnár et al., 2010*; *Barber et al., 2011*), the non-human primate (*Kauffman et al., 2021*) and 3D cellular models (*Tezera et al., 2020a*), and ultimately has been validated by patient registry studies (*Liu et al., 2022*; *Zhu et al., 2022*). Similarly, type II diabetes is associated with an increased risk of TB (*Dheda et al., 2016*), characterized by a hyper-inflammatory

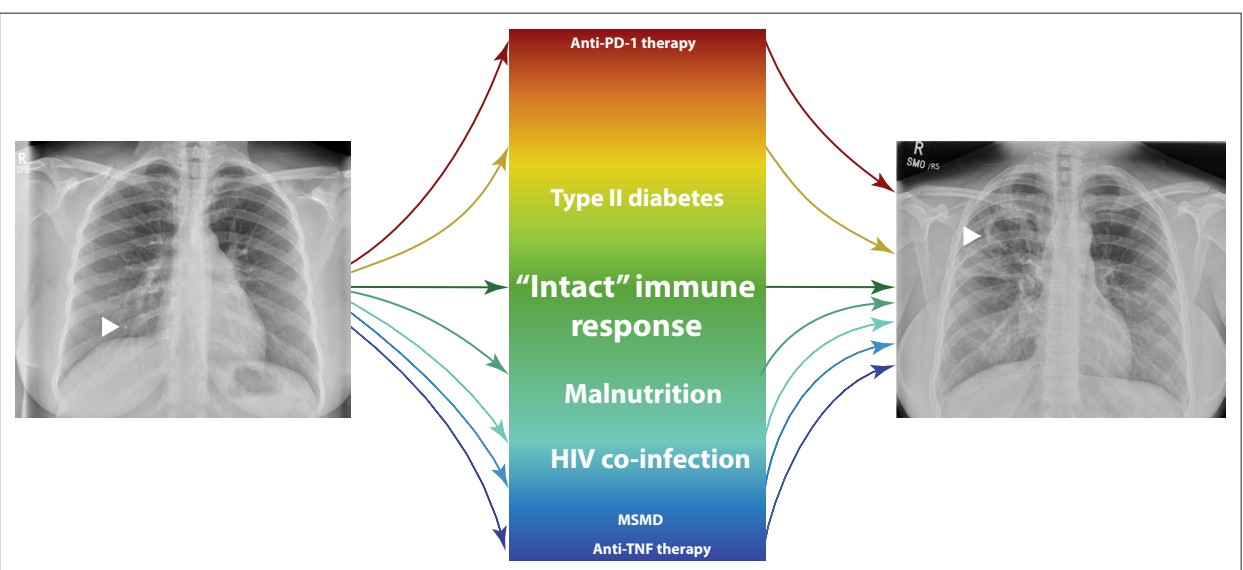

**Figure 1.** Multiple pathways can lead to tuberculosis (TB) disease from opposing immunological extremes. Generally, these can be viewed as immune deficiencies or immune excess, but the majority of patients who develop TB are relatively young with a competent immune response, illustrating the complexity of this spectrum. Though not quantitative, font size relates to contribution to global incidence. Left arrow: Ghon focus at lung base; Right arrow: cavity at lung apex.

immune response (*Eckold et al., 2021*). Therefore, diverse clinical evidence demonstrates that there are multiple immunological disturbances that can lead to TB disease (*Behr et al., 2024*; *Figure 1*).

Even when TB develops in the face of a 'normal' immune system, there are likely to be many different subgroups that have not yet been identified due to limitations in standard immunological profiling. A quarter of the world's population is thought to be exposed to Mtb (*Houben and Dodd, 2016*), and so the vast majority of those infected with Mtb remain healthy lifelong (*Trajman et al., 2025*), while those who progress to active TB likely represent distinct outlier populations. The diverse causes of active TB, such as anti-TNF, MSMD, HIV, diabetes, and anti-PD1, demonstrate that patients do not reach TB by a single pathway, but instead lose the immunological balance that controls Mtb by multiple paths. These observations may explain why genetic studies have generally failed to find consistent predispositions. Evidence of heritability can be demonstrated but is often hard to validate in different populations (*Schurz et al., 2024*). Mutations in the macrophage endosomal protein NRAMP1, for example, were shown to associate with TB in an early seminal study (*Bellamy et al., 1998*), but since then, few consistent traits of genetic susceptibility to TB have been identified (*Abel et al., 2014*).

Recent genomic studies demonstrate the complex co-evolution of host and Mtb. The International Tuberculosis Host Genetics Consortium's first analysis found one significant host genetic variant, human leukocyte antigen-II region (rs28383206), which conferred TB susceptibility across nine genome-wide association studies across three continents (*Schurz et al., 2024*). However, other variants previously associated with TB susceptibility were not replicated. Another approach using genome-to-genome analysis of paired human and Mtb samples in Peru identified another determinant on chromosome 6, rs3130660, in the flotillin-1 (FLOT1) gene (*Luo et al., 2024*). Together with our understanding of the co-adaptation of different Mtb lineages with human migrations (*Comas et al., 2013*), the host immune response to Mtb is likely to be dependent on ancestral-related genetic factors and complex host–pathogen dynamics which remain incompletely understood.

Studies of individuals who resist Mtb infection despite recurrent exposure generate a diverse list of potentially protective features, including T cell subsets or activation (*Cross et al., 2024*; *Sun et al., 2024b*; *Dallmann-Sauer et al., 2025*), TNF-α responses (*Simmons et al., 2022*), antibody activity (*Lu et al., 2019*), and innate immune training (*Verrall et al., 2020*). These human observations are supported by murine genetic experiments, which show that increased susceptibility to Mtb can result from a wide range of immunological alterations (*Smith et al., 2022*). Similarly, infant vaccine studies show that there are distinct patterns of response that may determine vaccine efficacy (*Fletcher et al., 2016*). However, despite this evidence of diversity, the majority of fundamental studies continue to seek a single underlying mechanism that leads to TB disease progression, which is incompatible with clinical observations.

Ultimately, to transmit efficiently, Mtb needs to cause pulmonary disease to then spread by airborne droplets (*Yoder et al., 2004*), and for Mtb, it does not matter the route taken, as long as it ends at pulmonary TB. Increasing evidence suggests that asymptomatic transmission may be important in high-incidence settings, potentially in the absence of overt pulmonary disease (*Ryckman et al., 2022*; *Dinkele et al., 2024*; *Patterson et al., 2024*). Experimentally, TB is a fundamentally challenging disease to model, as the interaction is prolonged and Mtb is an obligate human pathogen (*Elkington et al., 2019*). The primary driver of advances in immunology in the last decades has been transgenic mice (*Gros and Casanova, 2023*), but the mouse model of TB does not accurately reflect human disease (*Young, 2009*). Disease heterogeneity has been highlighted in describing clinical TB endotypes observed during active disease (*DiNardo et al., 2021*), with different endotypes exhibiting diverse immunological characteristics and association with outcome (*DiNardo et al., 2022*). However, we propose that insufficient attention has been given to the different immunological pathways that may converge on the same disease phenotype.

## Mtb's single successful establishment in humanity

Just as new tools are providing insights into the complexity of human TB progression, advances in mycobacterial genomics are highlighting the unique nature of the human-Mtb relationship (*Koleske et al., 2023*). Mtb is a near-clonal organism, with evidence suggesting that there has been only one successful and sustained penetration into the human population (*Comas et al., 2013*; *Koleske et al., 2023*; *Goig et al., 2025*). The entire spectrum of Mtb strains globally only differs by a total

of approximately 2000 SNPs (*Koleske et al., 2023*). This genetic conservation has persisted from the most recent common ancestor (*Chiner-Oms et al., 2019*) during the expansion of Mtb in human populations since the industrial revolution, which created much denser human aggregations suitable for transmission (*Dubos and Dubos, 1987*). This suggests that Mtb was already close to being optimized for human hosts. Further evolution may have been to increase transmission within specific populations as humans diverged genetically (*Goig et al., 2025*), with a recent study suggesting that Mtb may in fact be becoming attenuated to increase spread in populations (*Culviner et al., 2025*). Hence, Mtb may be evolving over time to optimize its transmission within humans depending on population density.

A very similar organism, *M. canetti*, can cause disease but cannot transmit from human to human (*Yenew et al., 2023*). Similarly, *M. bovis* is 99.9% identical to Mtb (*Garnier et al., 2003*), but has never achieved sustained human-to-human transmission, despite many millions of human exposures during the pre-pasteurization era and common lymph node infections (*Goig et al., 2025*). Therefore, human TB is caused exclusively by Mtb, unlike other very closely related mycobacteria, which maintain infection cycles in other higher mammals but not humans (*Goig et al., 2025*). Clues to the key pathogenic mechanisms may lie in the differences between mycobacterial species (*Danchuk et al., 2025*), but ultimately, similarities between Mtb strains must be critical to their ongoing success. For example, the hyper-conservation of T cell epitopes may be evidence that Mtb manipulates the host immune response to favor disease and transmission (*Comas et al., 2010*). Given the size of the Mtb genome, identifying the critical conserved features will be challenging, especially as half of its genes still lack a known function 25 years after it was first sequenced (*Nathan, 2023*).

One highly intriguing proposition is that latent TB may itself give humans an evolutionary advantage (*Nathan, 2023*). Exposure to Mtb modifies innate immune training via reprogramming hematopoietic stem cells (*Khan et al., 2020*), and so a mechanism whereby Mtb could protect from other fatal infections is plausible. Similarly, *M. bovis* BCG, a live-attenuated strain of *M. bovis*, causes innate immune training (*Kaufmann et al., 2018*), suggesting this effect is common to Mtb and BCG. BCG vaccination reduces mortality much more significantly than can be explained by the effect on TB incidence alone (*Jurczak and Druszczynska, 2025*), with a protective effect confirmed in diverse studies (*Moorlag et al., 2019*). For example, BCG reduces viral infections in infants in Africa (*Stensballe*

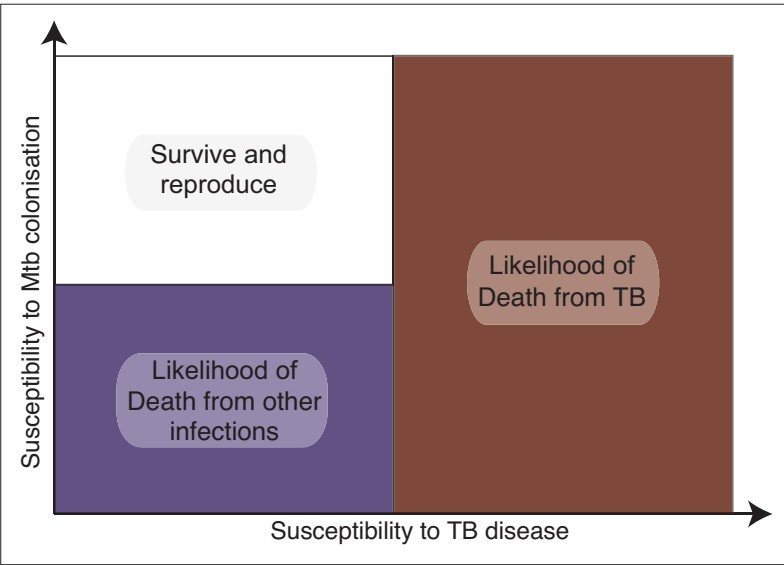

**Figure 2.** Selection pressure of prolonged co-evolution favors individuals permissive to asymptomatic *Mycobacterium tuberculosis* (Mtb) colonization but resistant to active disease. Over millennia, Mtb circulation in society will remove genetic traits that cause high susceptibility to active tuberculosis (TB) infection. Perhaps less intuitively, if Mtb generates trained immunity that protects against other fatal diseases, individuals with low susceptibility to initial Mtb infection will also be selected against due to increased mortality from other infections. The resulting population would then reflect modern humans: highly susceptible to initial Mtb colonization but with low susceptibility to TB disease. The figure illustrates the selective pressure concept, the increase in risk is not binary but gradual, with susceptibility determined by multiple aspects of the host immune response.

*et al., 2005*) and experimentally challenged adults (*Arts et al., 2018*), and associates with improved survival in Europe (*Rieckmann et al., 2017*). Mtb and BCG can protect against SARS-CoV2 infection (*Rosas Mejia et al., 2022*; *Hilligan et al., 2022*), although this did not translate to efficacy in a clinical trial (*Pittet et al., 2023*). Together, these observations strongly imply that mycobacterial infection protects from other infectious causes of death.

Mtb almost certainly first became established in humans in East Africa (*Comas et al., 2013*; *Goig et al., 2025*), potentially about 70,000 years ago (*Brites and Gagneux, 2015*), although this time-line is debated. Several human migrations out of Africa are thought to have preceded this date, but all ultimately became extinct, with the first sustained human dispersal 60–70,000 years ago (*Vallini et al., 2024*). Mtb diversity mirrors human mitochondrial genome diversity, further implying that Mtb disseminated with human populations from East Africa (*Comas et al., 2013*). The primary selective pressure in these early communities would have been infectious disease (*Dobson and Carper, 1996*). This raises a novel hypothesis that a survival advantage for the first successful human migrants out of Africa was Mtb circulating in the community, reducing mortality from other infectious diseases and thereby enabling sustainable population growth.

Mtb transmission may have benefitted humans by increasing innate immune resistance to infection at the cost of 10% disease penetrance that permits Mtb propagation. This selective pressure over many millennia would progressively remove genotypes that lead to high susceptibility to TB, but equally would select against individuals with complete resistance to initial Mtb infection (*Figure 2*). This could explain why consistent genetic traits for susceptibility or resistance to TB have been hard to identify (*Abel et al., 2014*). More recent mass infection events, such as the smallpox epidemics that killed approximately 25% of the population (*Dobson and Carper, 1996*), may have further favored individuals immunologically trained by Mtb. Likewise, successive waves of plague killed approximately 25% of the population, at the end of the Roman empire and then in the European middle ages, adding to selective pressure from endemic infections (*Little, 2007*; *Benedictow, 2004*). If humans have been selected to be permissive to Mtb infection but resistant to TB disease, which could be regarded as colonization, it suggests the development of active disease must be a relatively unusual event in a subset of outlier individuals. In some sense, we could be regarded as having a symbiotic relationship with Mtb, with disease representing a necessary evil, caused by an imbalance in the predominantly stable host–pathogen interaction (*Divangahi et al., 2018*; *Olive and Sassetti, 2018*).

## The host–pathogen interaction at a cellular level

The early histological era described the wide range of human TB lesions and granuloma types and identified TB as a disease characterized by spatial organization (*Hunter, 2016*). Classically, the granuloma has been proposed to be where the outcome of the host–pathogen interaction is determined (*Pagán and Ramakrishnan, 2018*). Recent methodological advances are permitting much greater dissection of events and further highlight the importance of spatial organization within the granuloma (*Sawyer et al., 2023*; *McCaffrey et al., 2022*; *Marakalala et al., 2016*). However, just as the early X-ray era showed some lesions progressing and some regressing, these studies demonstrate the great heterogeneity between granuloma types. Studies in the non-human primate have allowed investigation into features of progressing and controlling granulomas, identifying potential correlates of immune control (*Gideon et al., 2022*), but even this model only partially recapitulates human disease.

Furthermore, the recent spatial studies highlight the complexity of cellular players, including the established fulcrum of macrophages and T cells, but additionally the importance of B cells, neutrophils, and fibroblasts. For example, fibroblasts are emerging as important immune regulators in other lung diseases (*Ghonim et al., 2023*), and so it seems highly likely that they play an active role in TB-related inflammation. Fibroblast zonation can lead to feed-forward inflammatory loops and so may propagate disease (*Davidson et al., 2021*). Consequently, the full spectrum of cell types in Mtb-infected lesions needs to be considered. Given that multiple underlying immunological pathways can lead to active TB, it seems unlikely that a single cellular component will fully explain the balance between Mtb containment and progression to active disease. However, the majority of studies continue to look for a single consistent immune mechanism; discussions rarely state 'this is one of several potential routes to TB disease' (*Sun et al., 2024b*; *Gideon et al., 2022*; *Winchell et al., 2023*; *Swanson et al., 2023*; *Proulx et al., 2025*).

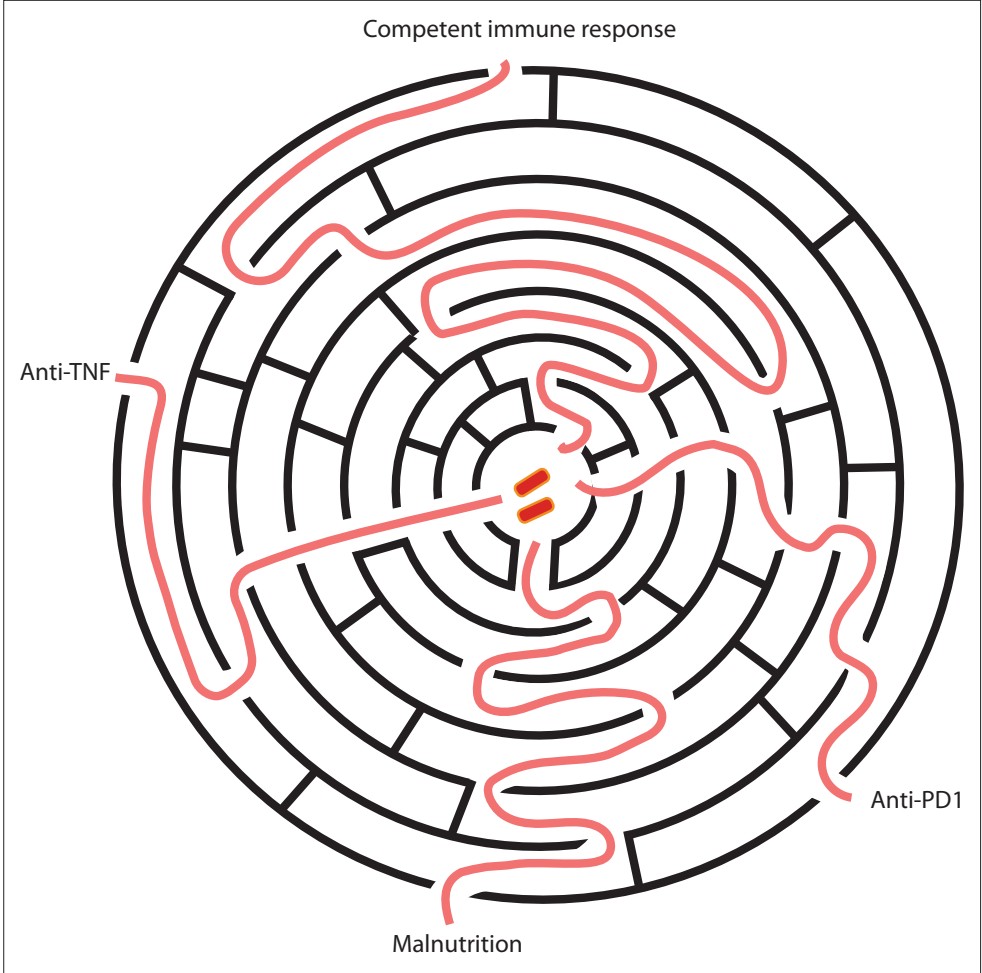

**Figure 3.** Schematic of the interactions needed for *Mycobacterium tuberculosis* (Mtb) to escape the host immune response. As control of Mtb requires a coordinated host response, there are multiple sequences of immune events that can ultimately result in progression to active tuberculosis (TB) disease. A major immune disturbance, such as TNF-α or PD-1 inhibition, gives a relatively direct pathway to active TB. However, most individuals develop TB due to a series of less apparent immune events and no clear global immune disturbance that can be identified by current immune profiling approaches.

Considering the clinical and experimental evidence, immunological failure is likely to be a multi-step process, whereby either one large deficit or numerous small imbalances can lead to progression and disease (*Figure 3*). Mtb and humans interact over many years, as the pathogen is difficult to eradicate due to its highly evolved survival mechanisms (*Russell, 2011*), and therefore in those individuals in whom it survives, there is a long period where it can escape host control. Potentially, these different paths may ultimately cross or converge; if so, understanding the key nodes will allow more broadly effective treatments to emerge. For example, lung extracellular matrix degradation could be regarded as a final common pathway (*Elkington et al., 2022*), but, again, this may result from different collagenases including macrophage-derived MMP-1 or neutrophil-derived MMP-8 (*Elkington et al., 2011*; *Ong et al., 2015*).

## Emerging methodologies and the challenge of data analysis

The recent adoption of 'omic' methodologies, including single-cell transcriptomics, spatial transcriptomics, and proteomics, offers unprecedented opportunities to dissect the mechanisms of human TB pathogenesis and accelerate the development of effective interventions. These approaches generate large-scale, complex datasets that capture the heterogeneity of TB lesions. However, this complexity

also presents significant analytical challenges. Standard computational pipelines, if not adapted to account for the biological and technical variability in these data, are unlikely to deliver robust or reproducible insights. A major obstacle is the integration of data from multiple studies and platforms, which can differ both within a single omic layer (horizontal integration) and across multiple omic layers (vertical integration) (*Zheng et al., 2024*). Such integration risks data loss and inconsistent results, especially when data are not harmonized.

The prolonged co-evolution between host and pathogen has resulted in multiple immunological pathways to active TB, significantly adding to the biological heterogeneity that complicates the analysis and interpretation of multi-omic data. Addressing this complexity requires well-annotated clinical cohorts that capture the full spectrum of TB heterogeneity, ideally with longitudinal outcome data rather than single time-point snapshots. Comprehensive clinical descriptors and associated data such as laboratory analyses and chest X-rays will permit definition of each clinical phenotype. To accommodate the diversity of disease pathways, novel bioinformatic approaches are needed that move beyond the assumption of a single sequence of events that results in active TB.

The power of multi-omic approaches to reveal complex molecular relationships depends on both the quality of the omic data and the fit between experimental design and data integration strategies. For example, correlation-based integration requires matched samples across omics, sufficient sample numbers, and comparable variance structures. These requirements are often overlooked, leading to insufficient power, noisy data, and unrealistic integration (*Tarazona et al., 2020*). Each omic technology brings its own challenges. Signal-to-noise ratios differ across modalities, and appropriate algorithms are needed to estimate sample size and power for each. Notably, using the same sample numbers across modalities does not ensure comparable statistical power; achieving equal power can require unbalanced sample sizes (*Tarazona et al., 2021*). Other recognized challenges include the interpretation and validation of multi-omic models, standardized annotation, and data sharing (*Tarazona et al., 2021*). Recent developments include web-based, user-friendly tools that enable both knowledge- and data-driven multi-omic integration (*Ewald et al., 2024*). Importantly, a wider use of artificial intelligence (AI) is transforming omics data analysis by providing robust methods to interpret complex biological datasets (*Ahmed et al., 2024*). Machine learning and deep learning techniques are now routinely applied to DNA, transcriptomic, proteomic, and metabolomic data, enabling more integrated and comprehensive analyses (*Ahmed et al., 2024*). In proteomics, for example, AI-driven approaches have enhanced peptide measurement predictions and accelerated biomarker discovery, often outperforming conventional assays (*Mann et al., 2021*). To improve interpretability, explainable AI (XAI) methods are increasingly employed, with feature relevance mapping and visual explanations emerging as preferred post hoc strategies (*Toussaint et al., 2023*). However, despite these developments, significant challenges remain in implementing XAI. Further research is needed to overcome these barriers and unlock the full translational potential of AI in omics analysis (*Mann et al., 2021*; *Toussaint et al., 2023*). Ultimately, the utility of these innovations will depend on the functional validation of the resulting models.

Given the complexity of human TB, careful study design and unbiased integration methods that accommodate data limitations are essential. Spatial context is particularly important, as TB pathology involves three-dimensional immune responses. Extracellular matrix remodeling is a hallmark of TB granulomas (*Elkington et al., 2022*), and the matrix regulates host cell biology (*Bansaccal et al., 2023*). Consequently, multiple data inputs such as matrix composition and organization may be needed alongside host and Mtb transcriptomic data. Ultimately, local cellular events must be modeled at the tissue level, providing a second layer of computational complexity (*Palla et al., 2022*).

Addressing these challenges will require substantial investment in analytical capacity. Multi-omics is already transforming our understanding of disease heterogeneity, facilitating the identification of previously unrecognized subgroups, refining prognostic and therapeutic approaches, and providing deeper mechanistic insights. This strategy has successfully stratified rare tumors (*Sun et al., 2024a*), profiled healthy populations (*Halama et al., 2024*), and enabled health screening to reveal previously hidden disease and risk subgroups (*Garg et al., 2024*), supporting the shift toward precision medicine. As human disease processes are rarely uniform, advances in multi-omic study design and analysis for TB will likely benefit a broad range of conditions. Ultimately, this comprehensive understanding of disease pathophysiology can then lead to more targeted and stratified treatment, although implementation will require developments in companion diagnostics to accurately stratify patients.

## Implications of diverse disease pathways for new TB interventions

Ultimately, the complexity of human TB underlies our inability to deploy a transformative intervention, and so global mortality remains depressingly high. The human–Mtb interaction is so closely co-evolved that experimental findings need to be interpreted in light of human disease phenomena. Multiomic studies in TB are currently being undertaken on small numbers due to cost and challenges in obtaining appropriate clinical samples, in specific regions, and so are unlikely to reflect the global heterogeneity. Therefore, results need to be interpreted with caution and wider studies will be needed to confirm the generalizability of findings. A critical aspect will be carefully curated and fully accessible metadata, so that as the body of omic datasets on human TB increases, they can be accurately integrated into wider analyses. Recurrent mining of these datasets is likely to be fundamental to understanding the breadth of TB pathogenesis. Missing metadata makes interpretation difficult, and in worst cases, misleading. In addition, innovative computational approaches will be required whereby the analysis specifically accommodates multiple immune pathways to disease.

Given the toll that TB takes in the poorest parts of the world, we have a moral imperative to end the epidemic (*Reid et al., 2023*). To achieve this, we must acknowledge the complexity of human TB that has resulted from our prolonged co-evolution with the pathogen and the selective pressure of persistent Mtb exposure over millennia. Success depends on integrating the full spectrum of TB disease into our bioinformatic analyses, and ultimately understanding TB's complexity can then inform logical interventions. If we seek a single mechanistic explanation of TB disease, this seems to be unlikely to be successful.

## Acknowledgements

PTE was supported by the MRC (MR/W025728/1), MTR by the Rosetrees Trust (CF-2021–2\126), SM by the MRC (MR/S024220/1), LBT by the Academy of Medical Sciences Springboard (SBF0010\1085), and AL by the Wellcome Trust (210662/Z/18/Z) and the BMGF (OPP1137006).

## Additional information

### Funding

| Funder | Grant reference number | Author |
|---|---|---|
| Medical Research Council | MR/W025728/1 | Paul T Elkington |
| Rosetrees Trust | CF-2021-2\126 | Michaela T Reichmann |
| Medical Research Council | MR/S024220/1 | Salah Mansour |
| Academy of Medical Sciences | SBF0010\1085 | Liku B Tezera |
| Wellcome Trust | 10.35802/210662 | Alasdair Leslie |
| Gates Foundation | OPP1137006 | Alasdair Leslie |

The funders had no role in study design, data collection and interpretation, or the decision to submit the work for publication. For the purpose of Open Access, the authors have applied a CC BY public copyright license to any Author Accepted Manuscript version arising from this submission.

### Author contributions

Michaela T Reichmann, Liku B Tezera, Laura Denney, Hannah Schiff, Andres Vallejo, Salah Mansour, Alasdair Leslie, Diana J Garay-Baquero, Conceptualization, Writing – review and editing; Paul T Elkington, Conceptualization, Funding acquisition, Writing – original draft

### Author ORCIDs

Michaela T Reichmann  https://orcid.org/0000-0002-6714-8400
Liku B Tezera  https://orcid.org/0000-0002-7898-6709

Salah Mansour https://orcid.org/0000-0002-5982-734X
Diana J Garay-Baquero https://orcid.org/0000-0002-9450-8504
Paul T Elkington https://orcid.org/0000-0003-0390-0613

Reviewer #1 (Public review): https://doi.org/10.7554/eLife.108175.3.sa1
Reviewer #2 (Public review): https://doi.org/10.7554/eLife.108175.3.sa2
Reviewer #3 (Public review): https://doi.org/10.7554/eLife.108175.3.sa3
Author response https://doi.org/10.7554/eLife.108175.3.sa4

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
