## [Editor Report · eLife Assessment]

This Review Article explores the intricate relationship between humans and *Mycobacterium tuberculosis* (Mtb), providing an additional perspective on tuberculosis (TB) disease. Specifically, this review focuses on the utilization of systems-level approaches to study TB, while highlighting challenges in the frameworks used to identify the relevant immunologic signals that may explain the clinical spectrum of disease. The work could be further enhanced by better defining key terms that anchor the review, such as ‘unified mechanism’ and ‘immunological route’. This review will be of interest to immunologists as well as those interested in evolution and host-pathogen interactions.

---

## [Referee Report · Reviewer #1 (Public review)]

Summary:

This is an interesting and useful review highlighting the complex pathways through which pulmonary colonisation or infection with *Mycobacterium tuberculosis* (Mtb) may progress to develop symptomatic disease and transmit the pathogen. I found the section on immune correlates associated with individuals who have clearly been exposed to and reacted to Mtb but did not develop latent infections particularly valuable. However, several aspects would benefit from clarification.

Strengths:

The main strengths lie in the arguments presented for a multiplicity of immune pathways to TB disease.

Weaknesses:

The main weaknesses lie in clarity, particularly in the precise meanings of the three figures.

I accept that there is a 'goldilocks zone' that underpins the majority of TB cases we see and predominantly reflects different patterns of immune response, but the analogies used need to be more clearly thought through.

---

## [Referee Report · Reviewer #2 (Public review)]

Summary:

This is a thought-provoking perspective by Reichmann et al, outlining supportive evidence that *Mycobacterium tuberculosis* co-evolved with its host *Homo sapiens* to both increase susceptibility to infection and reduce rates of fatal disease through decreased virulence. TB is an ancient disease where two modes of virulence are likely to have evolved through different stages of human evolution: one before the Neolithic Demographic Transition, where humans lived in sparse hunter-gatherer communities, which likely selected for prolonged Mtb infection with reduced virulence to allow for transmission across sparse populations. Conversely, following the agricultural and industrial revolutions, Mtb virulence is likely to have evolved to attack a higher number of susceptible individuals. These different disease modalities highlight the central idea that there are different immunological routes to TB disease, which converge on a disease phenotype characterized by high bacterial load and destruction of the extracellular matrix. The writing is very clear and provides a lot of supportive evidence from population studies and the recent clinical trials of novel TB vaccines, like M72 and H56. However, there are areas to support the thesis that have been described only in broad strokes, including the impact of host and Mtb genetic heterogeneity on this selection, and the alternative model that there are likely different TB diseases (as opposed to different routes to the same disease), as described by several groups advancing the concept of heterogeneous TB endotypes. I expand on specific points below.

Strengths:

(1) The idea that Mtb evolved to both increase transmission (and possible commensalism with humans) with low rates of reactivation is intriguing. The heterogeneous TB phenotypes in the collaborative cross model (PMID: 35112666) support this idea, where some genetic backgrounds can tolerate a high bacterial load with minimal pathology, while others show signs of pathogenesis with low bacterial loads. This supports the idea that the underlying host state, driven by a number of factors like genetics and nutrition, is likely to explain whether someone will co-exist with Mtb without pathology, or progress to disease. I particularly enjoyed the discussion of the protective advantages provided by Mtb infection, which may have rewired the human immune system to provide protection against heterologous pathogens- this is supported by recent studies showing that Mtb infection provides moderate protection against SARS-CoV-2 (PMID: 35325013, and 37720210), and may have applied to other viruses that are likely to have played a more significant role in the past in the natural selection of *Homo sapiens*.

(2) Modeling from Marcel Behr and colleagues (PMID: 31649096) indeed suggests that there are at least TB clinical phenotypes that likely mirror the two distinct phases of Mtb co-evolution with humans. Most of the TB disease progression occurs rapidly (within 1-2 years of exposure), and the rest are slow cases of reactivation over time. I enjoyed the discussion of the difference between the types of immune hits needed to progress to disease in the two scenarios, where you may need severe immune hits for rapid progression, a phenotype that likely evolved after the Neolithic transition to larger human populations. On the other hand, a series of milder immune events leading to reactivation after a long period of asymptomatic infection likely mirrors slow progression in the hunter-gatherer communities, to allow for prolonged transmission in scarce populations. Perhaps a clearer analysis of these models would be helpful for the reader.

Weaknesses:

(1) The discussion of genetic heterogeneity is limited and only discusses evidence from MSMD studies. Genetics is an important angle to consider in the co-evolution of Mtb and humans. There is a large body of literature on both host and Mtb genetic associations with TB disease. The very fact that host variants in one population do not necessarily cross-validate across populations is evidence in support of population-specific adaptations. Specific Mtb lineages are likely to have co-evolved with distinct human populations. A key reference is missing (PMID: 23995134), which shows that different lineages co-evolved with human migrations. Also, meta-analyses of human GWAS studies to define variants associated with TB are very relevant to the topic of co-evolution (e.g., PMID: 38224499). eQTL studies can also highlight genetic variants associated with regulating key immune genes involved in the response to TB. The authors do mention that Mtb itself is relatively clonal with ~2K SNPs marking Mtb variation, much of which has likely evolved under the selection pressure of modern antibiotics. However, some of this limited universe of variants can still explain co-adaptations between distinct Mtb lineages and different human populations, as shown recently in the co-evolution of lineage 2 with a variant common in Peruvians (PMID: 39613754).

(2) Although the examples of anti-TNF and anti-PD1 treatments are relevant as drivers of TB in limited clinical contexts, the bigger picture is that they highlight major distinct disease endotypes. These restricted examples show that TB can be driven by immune deficiency (as in the case of anti-TNF, HIV, and malnutrition) or hyperactivation (as in the case of anti-PD1 treatment), but there are still certainly many other routes leading to immune suppression or hyperactivation. Considering the idea of hyper-activation as a TB driver, the apparent higher rate of recurrence in the H56 trial referenced in the review is likely due to immune hyperactivation, especially in the context of residual bacteria in the lung. These different TB manifestations (immune suppression vs immune hyperactivation) mirror TB endotypes described by DiNardo et al (PMID: 35169026) from analysis of extensive transcriptomic data, which indicate that it's not merely different routes leading to the same final endpoint of clinical disease, but rather multiple different disease endpoints. A similar scenario is shown in the transcriptomic signatures underlying disease progression in BCG-vaccinated infants, where two distinct clusters mirrored the hyperactivation and immune suppression phenotypes (PMID: 27183822). A discussion of how to think about translating the extensive information from system biology into treatment stratification approaches, or adjunct host-directed therapies, would be helpful.

---

## [Referee Report · Reviewer #3 (Public review)]

Summary:

This perspective article by Reichmann et al. highlights the importance of moving beyond the search for a single, unified immune mechanism to explain host-Mtb interactions. Drawing from studies in immune profiling, host and bacterial genetics, the authors emphasize inconsistencies in the literature and argue for broader, more integrative models. Overall, the article is thought-provoking and well-articulated, raising a concept that is worth further exploration in the TB field.

Strengths:

Timely and relevant in the context of the rapidly expanding multi-omics datasets that provide unprecedented insights into host-Mtb interactions.

Weaknesses (Minor):

(1) Clarity on the notion of a "unified mechanism". It remains unclear whether prior studies explicitly proposed a single unifying immunological model. While inconsistencies in findings exist, they do not necessarily demonstrate that earlier work was uniformly "single-minded". Moreover, heterogeneity in TB has been recognized previously (PMIDs: 19855401, 28736436), which the authors could acknowledge.

(2) Evolutionary timeline and industrial-era framing. The evolutionary model is outdated. Ancient DNA studies place the Mtb's most recent common ancestor at ~6,000 years BP (PMIDs: 25141181; 25848958). The Industrial Revolution is cited as a driver of TB expansion, but this remains speculative without bacterial-genomics evidence and should be framed as a hypothesis. Additionally, the claim that Mtb genomes have been conserved only since the Industrial Revolution (lines 165-167) is inaccurate; conservation extends back to the MRCA (PMID: 31448322).

(3) Trained immunity and TB infection. The treatment of trained immunity is incomplete. While BCG vaccination is known to induce trained immunity (ref 59), revaccination does not provide sustained protection (ref 8), and importantly, Mtb infection itself can also impart trained immunity (PMID: 33125891). Including these nuances would strengthen the discussion.

---

## [Author Response]

The following is the authors’ response to the original reviews.

**eLife Assessment**
This Review Article explores the intricate relationship between humans and *Mycobacterium tuberculosis* (Mtb), providing an additional perspective on TB disease. Specifically, this review focuses on the utilization of systems-level approaches to study TB, while highlighting challenges in the frameworks used to identify the relevant immunologic signals that may explain the clinical spectrum of disease. The work could be further enhanced by better defining key terms that anchor the review, such as "unified mechanism" and "immunological route." This review will be of interest to immunologists as well as those interested in evolution and host-pathogen interactions.

We thank the editors for reviewing our article and for the primarily positive comments. We accept that better definition and terminology will improve the clarity of the message, and so have changed the wording as suggested above in the revised manuscript.

**Public Reviews:**

**Reviewer #1 (Public review):**
Summary:This is an interesting and useful review highlighting the complex pathways through which pulmonary colonisation or infection with *Mycobacterium tuberculosis* (Mtb) may progress to develop symptomatic disease and transmit the pathogen. I found the section on immune correlates associated with individuals who have clearly been exposed to and reacted to Mtb but did not develop latent infections particularly valuable. However, several aspects would benefit from clarification.Strengths:The main strengths lie in the arguments presented for a multiplicity of immune pathways to TB disease.Weaknesses:The main weaknesses lie in clarity, particularly in the precise meanings of the three figures.

We accept this point, and have completely changed figure 2, and have expanded the legends for figure 1 and 3 to maximise clarity.

I accept that there is a 'goldilocks zone' that underpins the majority of TB cases we see and predominantly reflects different patterns of immune response, but the analogies used need to be more clearly thought through.

We are glad the reviewer agrees with the fundamental argument of different patterns of immunity, and have revised the manuscript throughout where we feel the analogies could be clarified.

**Reviewer #2 (Public review):**
Summary:This is a thought-provoking perspective by Reichmann et al, outlining supportive evidence that *Mycobacterium tuberculosis* co-evolved with its host *Homo sapiens* to both increase susceptibility to infection and reduce rates of fatal disease through decreased virulence. TB is an ancient disease where two modes of virulence are likely to have evolved through different stages of human evolution: one before the Neolithic Demographic Transition, where humans lived in sparse hunter-gatherer communities, which likely selected for prolonged Mtb infection with reduced virulence to allow for transmission across sparse populations. Conversely, following the agricultural and industrial revolutions, Mtb virulence is likely to have evolved to attack a higher number of susceptible individuals. These different disease modalities highlight the central idea that there are different immunological routes to TB disease, which converge on a disease phenotype characterized by high bacterial load and destruction of the extracellular matrix. The writing is very clear and provides a lot of supportive evidence from population studies and the recent clinical trials of novel TB vaccines, like M72 and H56. However, there are areas to support the thesis that have been described only in broad strokes, including the impact of host and Mtb genetic heterogeneity on this selection, and the alternative model that there are likely different TB diseases (as opposed to different routes to the same disease), as described by several groups advancing the concept of heterogeneous TB endotypes. I expand on specific points below.Strengths:The idea that Mtb evolved to both increase transmission (and possible commensalism with humans) with low rates of reactivation is intriguing. The heterogeneous TB phenotypes in the collaborative cross model (PMID: 35112666) support this idea, where some genetic backgrounds can tolerate a high bacterial load with minimal pathology, while others show signs of pathogenesis with low bacterial loads. This supports the idea that the underlying host state, driven by a number of factors like genetics and nutrition, is likely to explain whether someone will co-exist with Mtb without pathology, or progress to disease. I particularly enjoyed the discussion of the protective advantages provided by Mtb infection, which may have rewired the human immune system to provide protection against heterologous pathogens- this is supported by recent studies showing that Mtb infection provides moderate protection against SARS-CoV-2 (PMID: 35325013, and 37720210), and may have applied to other viruses that are likely to have played a more significant role in the past in the natural selection of *Homo sapiens*.

We thank the reviewer for their positive comments, and also for pointing out work that we have overlooked citing previously. We now discuss and cite the work above as suggested

Modeling from Marcel Behr and colleagues (PMID: 31649096) indeed suggests that there are at least TB clinical phenotypes that likely mirror the two distinct phases of Mtb co-evolution with humans. Most of the TB disease progression occurs rapidly (within 1-2 years of exposure), and the rest are slow cases of reactivation over time. I enjoyed the discussion of the difference between the types of immune hits needed to progress to disease in the two scenarios, where you may need severe immune hits for rapid progression, a phenotype that likely evolved after the Neolithic transition to larger human populations. On the other hand, a series of milder immune events leading to reactivation after a long period of asymptomatic infection likely mirrors slow progression in the hunter-gatherer communities, to allow for prolonged transmission in scarce populations. Perhaps a clearer analysis of these models would be helpful for the reader.

We agree that we did not present these concepts in as much detail as we should, and so we now discuss this more on lines (81 – 83 and 184 - 187)

Weaknesses:The discussion of genetic heterogeneity is limited and only discusses evidence from MSMD studies. Genetics is an important angle to consider in the co-evolution of Mtb and humans. There is a large body of literature on both host and Mtb genetic associations with TB disease. The very fact that host variants in one population do not necessarily cross-validate across populations is evidence in support of population-specific adaptations. Specific Mtb lineages are likely to have co-evolved with distinct human populations. A key reference is missing (PMID: 23995134), which shows that different lineages co-evolved with human migrations. Also, meta-analyses of human GWAS studies to define variants associated with TB are very relevant to the topic of co-evolution (e.g., PMID: 38224499). eQTL studies can also highlight genetic variants associated with regulating key immune genes involved in the response to TB. The authors do mention that Mtb itself is relatively clonal with ~2K SNPs marking Mtb variation, much of which has likely evolved under the selection pressure of modern antibiotics. However, some of this limited universe of variants can still explain co-adaptations between distinct Mtb lineages and different human populations, as shown recently in the co-evolution of lineage 2 with a variant common in Peruvians (PMID: 39613754).

We thank the reviewer for these comments and agree we failed to cite and discuss the work from Sebastian Gagneux’s group on co-migration, which we now discuss. We include a new paragraph discussing co-evolution as suggested on lines 145 – 155 and 218 -220 , citing the work proposed, which we agree enhances the arguments about co-evolution.

Although the examples of anti-TNF and anti-PD1 treatments are relevant as drivers of TB in limited clinical contexts, the bigger picture is that they highlight major distinct disease endotypes. These restricted examples show that TB can be driven by immune deficiency (as in the case of anti-TNF, HIV, and malnutrition) or hyperactivation (as in the case of anti-PD1 treatment), but there are still certainly many other routes leading to immune suppression or hyperactivation. Considering the idea of hyper-activation as a TB driver, the apparent higher rate of recurrence in the H56 trial referenced in the review is likely due to immune hyperactivation, especially in the context of residual bacteria in the lung. These different TB manifestations (immune suppression vs immune hyperactivation) mirror TB endotypes described by DiNardo et al (PMID: 35169026) from analysis of extensive transcriptomic data, which indicate that it's not merely different routes leading to the same final endpoint of clinical disease, but rather multiple different disease endpoints. A similar scenario is shown in the transcriptomic signatures underlying disease progression in BCG-vaccinated infants, where two distinct clusters mirrored the hyperactivation and immune suppression phenotypes (PMID: 27183822). A discussion of how to think about translating the extensive information from system biology into treatment stratification approaches, or adjunct host-directed therapies, would be helpful.

We agree with the points made and that the two publications above further enhance the paper. We have added discussion of the different disease endpoints on line 65 - 67, the evidence regarding immune herpeactivation versus suppression in the vaccination study on lines 162 - 164, and expanded on the translational implications on lines 349 – 352.

**Reviewer #3 (Public review):**
Summary:This perspective article by Reichmann et al. highlights the importance of moving beyond the search for a single, unified immune mechanism to explain host-Mtb interactions. Drawing from studies in immune profiling, host and bacterial genetics, the authors emphasize inconsistencies in the literature and argue for broader, more integrative models. Overall, the article is thought-provoking and well-articulated, raising a concept that is worth further exploration in the TB field.Strengths:Timely and relevant in the context of the rapidly expanding multi-omics datasets that provide unprecedented insights into host-Mtb interactions.Weaknesses (Minor):Clarity on the notion of a "unified mechanism". It remains unclear whether prior studies explicitly proposed a single unifying immunological model. While inconsistencies in findings exist, they do not necessarily demonstrate that earlier work was uniformly "single-minded". Moreover, heterogeneity in TB has been recognized previously (PMIDs: 19855401, 28736436), which the authors could acknowledge.

We accept this point and have toned down the language, acknowledging that we are expanding on an argument that others have made, whilst focusing on the implications for the systems immunology era, and cite the previous work as suggested.

Evolutionary timeline and industrial-era framing. The evolutionary model is outdated. Ancient DNA studies place the Mtb's most recent common ancestor at ~6,000 years BP (PMIDs: 25141181; 25848958). The Industrial Revolution is cited as a driver of TB expansion, but this remains speculative without bacterial-genomics evidence and should be framed as a hypothesis. Additionally, the claim that Mtb genomes have been conserved only since the Industrial Revolution (lines 165-167) is inaccurate; conservation extends back to the MRCA (PMID: 31448322).

Our understanding is that the evolutionary timeline is not fully resolved, with conflicting evidence proposing different dates. The ancient DNA studies giving a timeline of 6,000 years seem to oppose the evidence of evidence of Mtb infection of humans in the middle east 10,000 years ago, and other estimates suggesting 70,000 years. Therefore, we have cited the work above and added a sentence highlighting that different studies propose different timelines. We would propose the industrial revolution created the ideal societal conditions for the expansion of TB, and this would seem widely accepted in the field, but have added a proviso as suggested. We did not intent to claim that Mtb genomes have been conserved since the industrial revolution, the point we were making is that despite rapid expansion within human populations, it has still remained conserved. We therefore have revised our discussion of the conservation of the Mtb genomes on lines and 72 – 74, 81 – 83 and 185 – 190.

Trained immunity and TB infection. The treatment of trained immunity is incomplete. While BCG vaccination is known to induce trained immunity (ref 59), revaccination does not provide sustained protection (ref 8), and importantly, Mtb infection itself can also impart trained immunity (PMID: 33125891). Including these nuances would strengthen the discussion.

We have refined this section. We did cite PMID: 33125891 in the original submission but have changed the wording to emphasise the point on line …

**Recommendations for the authors:**

**Reviewer #1 (Recommendations for the authors):**
AbstractLine 30: What is an immunological route? Suggest”...host-pathogen interaction, with diverse immunological processes leading to TB disease (10%) or stable lifelong association or elimination. We suggest these alternate relationships result from the prolonged co-evolution of the pathogen with humans and may even confer a survival advantage in the 90% of exposures that do not progress to disease.”

Thank you, we have reworded the abstract along the lines suggested above, but not identically to allow for other reviewer comments.

IntroductionLn 43: It is misleading to suggest that the study of TB was the leading influence in establishing the Koch's postulates framework. Many other infections were involved, and Jacob Henle, one of Koch's teachers, is credited with the first clear formulation (see Evans AS. 1976 THE YALE JOURNAL OF BIOLOGY AND MEDICIN PMID: 782050).

We have downplayed the language, stating that TB “contributed” to the formulation if Koch’s postulated.

Ln 46: While the review rightly emphasises intracellular infection in macrophages, the importance and abundance of extracellular bacilli should not be ignored, particularly in transmission and in cavities.

We agree, and have added text on the importance of extracellular bacteria and transmission.

Ln: 56: This is misleading as primary disease prevention is implied, whereas the vaccine was given to individuals presumed to be already infected (TST or IGRA positive). Suggest ..."reduces by 50% progression to overt TB disease when given to those with immunological evidence of latent infection.

Thank you, edit made as suggested

Ln 62: Not sure why it is urgent. Suggest "high priority".

Wording changed as suggested.

Figure 1 needs clarification. The colour scale appears to signify the strength or vigour of the immune response so that disease is associated with high (orange/red) or low (green/blue) activity. The arrows seem to imply either a sequence or a route map when all we really have is an association with a plausible mechanistic link. They might also be taken to imply a hierarchy that is not appropriate. I'm not sure that the X-rays and arrows add anything, and the rectangle provides the key information on its own. Clarify please.

We have clarified the figure legend. We feel the X-rays give the clinical context, and so have kept them, and now state in the legend that this is highlighting that there are diverse pathways leading to active disease to try to emphasise the point the figure is illustrating.

Ln 149-157: I agree that the current dogma is that overt pulmonary disease is required to spread Mtb and fuel disease prevalence. It is vitally important to distinguish the spread of the organism from the occurrence of disease (which does not, of itself, spread). However, both epidemiological (e.g. Ryckman TS, et al. 2022Proc Natl Acad Sci U S A:10.1073/pnas.2211045119) and recent mechanistic (Dinkele R, et al. 2024iScience:10.1016/j.isci.2024.110731, Patterson B, et al. 2024Proc Natl Acad Sci U S A:10. E1073/pnas.2314813121, Warner DF, et al. 2025Nat Rev Microbiol:10.1038/s41579-025-01201-x) studies indicate the importance of asymptomatic infections, and those associated with sputum positivity have recently been recognised by WHO. I think it will be important to acknowledge the importance of this aspect and consider how immune responses may or may not contribute.I regard the view that Mtb is an obligate pathogen, dependent on overt pTB for transmission, as needing to be reviewed.

We agree that we did not give sufficient emphasis to the emerging evidence on asymptomatic infections, and that this may play an important part in transmission in high incidence settings. We now include a discussion on this, and citation of the papers above, on lines 168 – 170.

Ln 159: The terms colonise and colonisation are used, without a clear definition, several times. My view is that both refer to the establishment and replication of an organism on or within a host without associated damage. Where there is associated damage, this is often mediated by immune responses. In this header, I think "establishment in humanity" would be appropriate.

We agree with this point and have changed the header as suggested, and clarified our meaning when we use the term colonisation, which the reviewer correctly interprets.

Ln 181-: I strongly support the view that Mtb has contributed to human selection, even to the suggestion that humanity is adapted to maintain a long-term relationship with Mtb

Thank you, and we have expanded on this evidence as suggested by other reviewers.

Ln 189: improved.

Apologies, typo corrected.

Figure 2: I was also confused by this. The x-axis does not make sense, as a single property should increase. Moreover, does incidence refer to incidence in individuals with that specific balance of resistance and susceptibility, or contribution to overall global incidence - I suspect the latter (also, prevalence would make more sense). At the same time, the legend implies that those with high resistance to colonisation will be infrequent in the population, suggesting that the Y axis should be labelled "frequency in human population". Finally, I can't see what single label could apply to the X axis. While the implication that the majority of global infections reflect a balance between the resistance and susceptibilities is indicated, a frequency distribution does not seem an appropriate representation.

The reviewer is correct that the X axis is aiming to represent two variables, which is not logical, and so we have completely changed this figure to a simple one that we hope makes the point clearly and have amended the legend appropriately. We are aiming to highlight the selective pressures of Mtb on the human population over millennia.

Ln 244: Immunological failure - I agree with the statement but again find the figure (3) unhelpful. Do we start or end in the middle? Is the disease the outside - if so, why are different locations implied? The notion of a maze has some value, but the bacteria should start and finish in the same place by different routes.

We are attempting to illustrate the concept that escape from host immunological control can occur through different mechanisms. As this comment was just from one reviewer, we have left the figure unchanged but have expanded the legend to try to make the point that this is just a conceptual illustration of multiple routes to disease.

Ln 262 onward: I broadly agree with the points made about omic technologies, but would wish to see major emphasis on clear phenotyping of cases. There is something of a contradiction in the review between the emphasis on the multiplicity of immunological processes leading ultimately to disease and the recommendation to analyse via omics, which, in their most widely applied format, bundle these complexities into analyses of the humoral and cellular samples available in blood. Admittedly, the authors point out opportunities for 3-dimensional and single-cell analyses, but it is difficult to see where these end without extrapolation ad infinitum.

We totally agree that clear phenotyping of infection is critical, and expand on this further on lines 307 - 309.

**Reviewer #2 (Recommendations for the authors):**
I suggest expanding on the genetic determinants of Mtb/host co-evolution.

Thank you, we have now expanded on these sections as suggested.

**Reviewer #3 (Recommendations for the authors):**
We are in an era of exploding large-scale datasets from multi-omics profiling of Mtb and host interactions, offering an unprecedented lens to understand the complexity of the host immune response to Mtb-a pathogen that has infected human populations for thousands of years. The guiding philosophy for how to interpret this tremendous volume of data and what models can be built from it will be critical. In this context, the perspective article by Reichmann et al. raises an interesting concept: to "avoid unified immune mechanisms" when attempting to understand the immunology underpinning host-Mtb interactions. To support their arguments, the authors review studies and provide evidence from immune profiling, host and bacterial genetics, and showcase several inconsistencies. Overall, this perspective article is well articulated, and the concept is worthwhile for further exploration. A few comments for consideration:Clarity on the notion of a "unified mechanism". Was there ever a single, clearly proposed unified immunological mechanism? For example, in lines 64-65, the authors criticize that almost all investigations into immune responses to Mtb are based on the premise that a unifying disease mechanism exists. However, after reading the article, it was not clear to me how previous studies attempted to unify the model or what that unifying mechanism was. While inconsistencies in findings certainly exist, they do not necessarily indicate that prior work was guided by a unified framework. I agree that interpreting and exploring data from a broader perspective is valuable, but I am not fully convinced that previous studies were uniformly "single-minded". In fact, the concept of heterogeneity in TB has been previously discussed (e.g., PMIDs: 19855401, 28736436).

We accept this point, and that we have overstated the argument and not acknowledged previous work sufficiently. We now downplay the language and cite the work as proposed.

However, we would propose that essentially all published studies imply that single mechanisms underly development of disease. The authors are not aware of any manuscript that concludes “Therefore, xxxx pathway is one of several that can lead to TB disease”, instead they state “Therefore, xxxx pathway leads to TB disease”. The implication of this language is that the mechanism described occurs in all patients, whilst in fact it likely only is involved in a subset. We have toned down the language and expand on this concept on line 268 – 270.

Evolutionary timeline and industrial-era framing. The evolutionary model needs updating. The manuscript cites a "70,000-year" origin for Mtb, but ancient-DNA studies place the most recent common ancestor at ~6,000 years BP (PMIDs: 25141181; 25848958). The Industrial Revolution is invoked multiple times as a driver of TB expansion, yet the magnitude of its contribution remains debated and, to my knowledge, lacks direct bacterial-genomics evidence for causal attribution; this should be framed as a hypothesis rather than a conclusion. In addition, the statement in lines 165-167 is inaccurate: at the genome level, Mtb has remained highly conserved since its most recent common ancestor-not specifically since the Industrial Revolution (PMID: 31448322).

We accept these points and have made the suggested amendments, as outlined in the public responses. Our understanding is that the evidence about the most common ancestor is controversial; if the divergence of human populations occurred concurrently with Mtb, then this must have been significantly earlier than 6,000 years ago, and so there are conflicting arguments in this domain.

Trained immunity and TB infection. The discussion of trained immunity could be expanded. Reference 59 suggests the induction of innate immune training, but reference 8 reports that revaccination does not confer protection against sustained TB infection, indicating that at least "re"-vaccination may not enhance protection. Furthermore, while BCG is often highlighted as a prototypical inducer of trained immunity, real-world infection occurs through Mtb itself. Importantly, a later study demonstrated that Mtb infection can also impart trained immunity (PMID: 33125891). Integrating these findings would provide a more nuanced view of how both vaccination and infection shape innate immune training in the TB context.

We thank the reviewer for these suggestions and have edited the relevant section to include these studies.